# Net Heart Rate for Estimating Oxygen Consumption in Active Adults

**DOI:** 10.3390/jfmk9020066

**Published:** 2024-04-07

**Authors:** José A. Bragada, Pedro M. Magalhães, Eric São-Pedro, Raul F. Bartolomeu, Jorge E. Morais

**Affiliations:** 1Department of Sport Sciences, Instituto Politécnico de Bragança, 5300-252 Bragança, Portugal; jbragada@ipb.pt (J.A.B.); pmaga@ipb.pt (P.M.M.); eric.pedro@ipb.pt (E.S.-P.); bartolomeu@ipg.pt (R.F.B.); 2Research Centre for Active Living and Wellbeing (LiveWell), Instituto Politécnico de Bragança, 5300-252 Bragança, Portugal; 3Department of Sport Sciences, Polytechnic of Guarda, 6300-559 Guarda, Portugal; 4Sport Physical Activity and Health Research & Innovation Center (SPRINT), 2040-413 Rio Maior, Portugal

**Keywords:** net heart rate, oxygen consumption, walk, run, cycling, StepTest4all

## Abstract

The aim of this study was to verify the accuracy of predicting oxygen consumption (O_2_) in predominantly aerobic activities based on net heart rate (netHR), sex, and body mass index (BMI) in active adults. NetHR is the value of the difference between the resting HR (HR_rest_) and the average HR value obtained during a given session or period of physical activity. These activities must be continuous, submaximal, and of a stabilized intensity. The magnitude of the netHR depends mainly on the intensity of the exercise. The HR is measured in beats per minute (bpm). A total of 156 participants, 52 women and 104 men, between the ages of 18 and 81, had their netHR and net oxygen intake (netVO_2_) assessed. There were 79 participants in group 1 (prediction sample) (52 males and 27 females). There were 77 people in group 2 (validation sample) (52 males and 25 females). The results of the multiple linear regression showed that netVO_2_ (R^2^ = 85.2%, SEE = 3.38) could be significantly predicted by sex (*p* < 0.001), netHR (*p* < 0.001), and BMI (*p* < 0.001). The Bland–Altman plots satisfied the agreement requirements, and the comparison of the measured and estimated netVO_2_ revealed non-significant differences with a trivial effect size. We calculated the formula NetVO_2_ (mL/(kg·min)) = 16 + 3.67 (sex) + 0.27 (netHR) − 0.57 (BMI) to predict netVO_2_, where netVO_2_ is the amount of oxygen uptake (mL/(kg·min)) above the resting value, netHR is the heart rate (beats per minute) above the resting value measured during exercise, sex is equal to zero for women and one for men, and BMI is the body mass index. In addition, based on the knowledge of VO_2_, it was possible to estimate the energy expenditure from a particular training session, and to determine or prescribe the exercise intensity in MET (metabolic equivalent of task).

## 1. Introduction

Measurements of oxygen consumption (VO_2_) are frequently made during health-related exercise or athletic training, particularly during medium-to-long-duration activities. Determining VO_2_ is crucial in activities where aerobic metabolism serves as the primary energy source for two main reasons: (i) it enables the evaluation of exercise intensity leading to more rigorous and customized training control, and (ii) it allows the estimation of energy expenditure (EE, in kilocalories or kilojoules) during steady-state efforts [1].

Direct spirometry is one method of measuring VO_2_ [1]. There are other methods for this purpose, such as estimation based on exercise performance or heart rate (HR) readings [2,3,4,5,6]. In the case of direct spirometry, it is assumed that a change in effort is directly proportional to an increase in HR and VO_2_ [7]. It is generally accepted that the variation in HR percentage in relation to maximal heart rate (HR_max_) or reserve heart rate (HR_reserve_) closely follows the variation of VO_2_ in relation to maximal oxygen consumption (VO_2max_) in each subject. Nonetheless, there are circumstances in which determining HR_max_ is neither feasible nor sufficient, and estimating it using formulas does not always seem to be suitable, particularly in older adults [8]. For instance, it is usually not advisable to determine HR_max_ in older adults, those with physical restrictions, or those with unique characteristics.

Net heart rate (netHR) is the value of the difference between resting HR (HR_rest_) and the average HR value obtained during a given session or period of physical activity. These activities must be continuous, submaximal, and of a stabilized intensity. The magnitude of the netHR depends mainly on the intensity of the exercise. Some prior research appears to support the applicability of this method [9,10,11].

Given the scarcity of research relating the variation in netHR to the variation in net oxygen consumption (netVO_2_), it would be interesting to confirm whether these two variables are directly proportional. If so, one variable could then be used to predict the other. Confirmation of the close relationship between netHR and netVO_2_ could even make it possible to estimate the EE during physical activities in a relatively simple way, based on determining the amount of oxygen expended in each physical activity or exercise. Knowing the number of liters of oxygen expended in submaximal, continuous, and steady-state activities, it is possible to determine the EE of a given physical activity by multiplying the number of liters of VO_2_ consumed by five. This gives us the energy expenditure in kcal [1,12].

There are some procedures that allow for the estimation of VO_2_ from the HR [13,14,15]. However, these estimations do not take HR_rest_ into consideration. HR_rest_ is a determining variable that can influence such estimations because this value does not depend on the activity performed. This is because the HR value measured during each activity integrates the HR_rest_ value, so it should not be associated directly with the EE inherent to the practice of a given exercise. Thus, it seems more appropriate to associate exercise intensity with netHR. In fact, the variation in intensity causes changes in HR magnitude above the resting value.

Usually, the estimation of VO_2_ from HR values is complemented by other variables that make the estimation more accurate. Among these variables, age, sex, and body mass index (BMI) are the most common [16,17]. In addition, the use of netHR can be advantageous in the evaluation and prescription of physical activity intensity since it does not imply knowledge of HR_max_. The use of net values is justified when one intends to study the relationship between HR, VO_2_, and EE and the level of intensity of the physical activity. Resting values exist even during complete inactivity of an individual. As a result, the measured values for those variables always line up with the total of the values added by engaging in physical activity and the resting values. NetHR and netVO_2_ in this study refer to the extent to which those variables deviate from resting values.

The main objective of this research was to validate a procedure for estimating netVO_2_ based on netHR, in combination with other variables, specifically sex and BMI, during physical activity or exercise (treadmill, cycling, and step exercises). The complementary objective was to demonstrate that it is possible to estimate the energy expenditure in a given training session from the netHR and the duration of the exercise.

## 2. Materials and Methods

### 2.1. Global Study Design

This study aimed to investigate the possibility of using netHR, in combination with other variables, to estimate EE during three types of physical activity (or exercise) that are widely used in cardiovascular capacity training (walking/running; cycling; and climbing/climbing steps). We used three ergometers to collect the data: a treadmill, cycle ergometer, and step ergometer. The protocols are described below. The main variables collected were anthropometric measurements, body composition, and the heart rate and respective exercise intensity values. Figure 1 below summarizes the data collection procedures.

### 2.2. Subjects and Samples

The sample consisted of 156 adults aged between 18 and 81 years, of whom, 52 were women and 104 were men. For each age group, the distribution was as follows: 18–29 years old: 72 people (62 men and 10 women); 30–59 years old: 33 people (18 men and 15 women); and more than 60 years old: 51 people (24 men and 27 women). Table 1 presents the demographics of all the subjects.

Of the 156 subjects in the whole sample, 79 performed a protocol on a treadmill, 40 on a cycle ergometer, and 37 on a step test. Since everyone performed at several levels of intensity, the total number of data points analyzed was 689 (prediction group: 351; validation group: 338). The whole sample was randomly split into two groups: group 1 (prediction group), which comprised 79 subjects (52 males and 27 females) and 351 datapoints, and group 2 (validation group), which comprised 77 subjects (52 males and 25 females) and 338 datapoints. Table 2 presents the comparison between groups in all the demographic variables measured.

All subjects signed a document to give permission to proceed with the evaluations, and the study was approved by the ethics committee of the Polytechnic Institute of Bragança, Portugal (N.º 120/2022).

### 2.3. Data Collection

Height was measured to the nearest 0.1 cm using a digital stadiometer (Seca, Model 242, Hamburg, Germany). Body mass was measured to the nearest 0.1 kg on an electronic scale (Seca, 884, Hamburg, Germany).

Data were acquired for the three types of exercise: walk/run on a treadmill; cycle ergometer; and step-ups and downs (StepTest4all). The tests were performed on a treadmill (45% of the data), a cyclo-ergometer (26.4% of the data), and a step (28.6% of the data). In each situation, the tests were progressive and continuous until 80% of the estimated HR_max_ was reaching according to the formula HR = 208 − 0.7 × Age [18] or until the subject requested to stop the test. This situation never happened, but it was a possibility to consider.

VO_2_ and HR were measured continuously using a stationary breath-by-breath electronic metabolic device (Cortex, Model MetaLyzer 3B, Leipzig, Germany). The device includes a heart rate transmitter (Polar Electro Oy, Kempele, Finland). The apparatus was calibrated with standard gases before each test. According to the manufacturer’s manual, the standard error is 0.1% for the O_2_ and CO_2_ sensors.

The walking/running activities were performed on a treadmill (Woodway, model 55 Sport, Germany). Cycling activities were performed with a bicycle ergo-trainer (Tacx Bushido Smart T2780, Wassenaar, The Netherlands). The resistance was dynamically controlled; it was constantly calculated and adjusted to maintain the previously determined power level in watts. The step test followed a specific protocol: StepTest4all (detailed in [19]).

### 2.4. Resting Heart Rate and Resting Oxygen Consumption

HR_rest_ and VO_2rest_ were measured before each test. The subject remained seated for 10 min, as relaxed as possible in a quiet environment, with a temperature between 20 and 22 degrees Celsius and reduced movements. HR and VO_2_ were measured during the entire resting period. The average values for the last two minutes were deemed the resting values. VO_2_ and HR were treated individually based on intensity level. Each participant only performed one protocol. Therefore, for each subject, several data points were considered (as many as the number of levels performed).

### 2.5. Treadmill Protocol

The subjects visited the laboratory twice. On the first visit, the subjects walked on the treadmill to become used to it. On the second day, they performed the protocol. VO_2_ and HR were measured continuously for each subject while performing the following activities in sequence: walk on a treadmill at 3 km/h, walk at 4.5 km/h, and walk/run at 6 or 7.5 km/h. During all steps, the VO_2_ and heart rate were collected for 6 min, and the mean value of the last minute was used for the data analysis. The test ended when the intensity reached a threshold of 80% of the estimated HR_max_ or if the subject asked for it to end. These subjects, although active, had no experience with high-intensity training. Thus, the researchers decided not to take risks regarding uninhibited efforts. In addition, some subjects were over 50 years of age, thus performing a maximum test without the presence of a doctor was not advisable.

### 2.6. Cyclo-Ergometer Protocol

VO_2_ and HR were measured continuously for each subject while cycling on the electronic trainer (Tacx, Bushido). The subject pedaled on the cycle ergometer for 4 to 6 steps for 4 min each. The power in the first step was 100 W, and it increased to 30 W for the next level. The test ended when the HR at the end of the step reached values equal to or above 80% of the estimated HR_max_ or if the subject asked for it to end. The values obtained in the last minute of each level were considered for further analysis.

### 2.7. StepTest4all Protocol

The evaluation in the step test was performed through continuous assessments of HR and VO_2_. The subjects performed 4 to 6 steps of increasing intensity. The height of the step was estimated based on the subjects’ characteristics, ranging from 20 to 40 cm. The step height was set considering the sex and height of the subject, their body mass index, and their level of physical activity. The pace was defined by a metronome. The pace of the first step was 12.5 cycles per minute and progressed by 2.5 cycles per minute after each minute. For more details, see [19]. The values obtained in the last five seconds of each step were considered for further analysis. The test ended when the intensity reached a threshold of 80% of the estimated HR_max_ or if the subject asked for it to end.

### 2.8. Statistical Analysis

Initially, Kolmogorov–Smirnov and Levene tests were used to assess normality and homoscedasticity, respectively. Descriptive data means and one standard deviation (1 SD) were calculated. For the prediction of netVO_2_, a stepwise regression (backward elimination) was computed with the inclusion of all the variables in the study (sex, body mass, height, BMI, and netHR). The final model retained only significant predictors (*p* < 0.05).

The comparison between the measured and estimated netVO_2_ values included two procedures: (i) data comparison between the mean measured and estimated VO_2_ values and (ii) a Bland–Altman analysis. For the data comparison, the paired sample t-test (*p* < 0.05) was used to compare the measured and estimated netVO_2_ values. The mean difference, 95% confidence intervals, and Cohen’s d as the effect size index were used. Cohen’s d was deemed (i) trivial if 0 ≤ d < 0.20; (ii) small if 0.20 ≤ d < 0.60; (iii) moderate if 0.60 ≤ d < 1.20; (iv) large if 1.20 ≤ d < 2.00; (v) very large if 2.00 ≤ d < 4.00; and (vi) nearly distinct if d ≥ 4.00 [20]. The Bland–Altman analysis generated plots of the difference and average of the measured and estimated netVO_2_ values [21]. The qualitative assessment found that at least 80% of the plots were within ± 1.96 standard deviation of the difference.

## 3. Results

The most significant outcome was that it is possible to meaningfully estimate the netVO_2_ associated with a specific physical activity based on netHR (*p* < 0.001), sex (*p* < 0.001), and BMI (*p* < 0.001). The multiple linear regression retained sex (*p* < 0.001), netHR (*p* < 0.001), and BMI (*p* < 0.001) as significant predictors of netVO_2_ (R^2^ = 85.2%, SEE = 3.38). The prediction equation can be expressed as:(1)$${netVO}_{2}=16.006+3.619\cdot (Sex)+0.269\cdot (netHR)-0.569\cdot \left(BMI\right)$$
where netVO_2_ corresponds to the oxygen uptake (mL/(kg·min)) above the resting value, netHR corresponds to the heart rate (beats per minute) above HR_rest_, sex corresponds to a value of zero for women or 1 for men, and BMI is the body mass index (kg/m^2^). Table 3 presents the comparison between the measured and estimated netVO_2_ values. The results showed non-significant differences with a trivial effect size.

Figure 2 illustrates the Bland–Altman plots which met the agreement criteria, with over 80% of the plots falling within the 95% CI. Specifically, nearly all the data points were within the 95% CI.

## 4. Discussion

The main purpose of this study was to estimate the netVO_2_ based on netHR, in combination with other variables. The prediction model of netVO_2_ retained the netHR, sex, and BMI variables. However, it is important to note that in the same individual, BMI changes slowly over time and sex remains constant. Thus, one should indicate that the most influential variable in the estimation of netVO_2_ is netHR.

Estimating the VO_2_ associated with physical activity and exercise is possible using several methods [22,23,24]. The most rigorous methods involve assessing respiratory exchange during exercise, especially VO_2_, carbon dioxide consumption, and the respiratory quotient [1]. There are other ways of estimating VO_2_ from formulas based on different types of exercises that can be applied to different populations and age groups [25,26,27,28,29]. The formulas proposed by the American College of Sport Medicine (ACSM), based on the intensity of the exercise and the subject’s own characteristics, are recognized and universally accepted [7]. It is also possible to estimate VO_2_ from HRreserve [30]. All these procedures have advantages and disadvantages. The main advantages include accessibility, non-invasiveness, low equipment costs, and the possibility of continuous monitoring. Some of the commonly cited limitations include the accuracy of predictions, which is affected by both external and internal influences on the individual. Factors such as age, physical condition, altitude, and hydration level can influence the relationship between heart rate and EE during physical exercise [1]. Furthermore, as mentioned in a review article [31], heart rate values may vary for the same exercise and the same person due to other reasons, such as hydration level, ambient temperature, the time of day, whether it is a weekday or weekend day, or even stress [32].

Despite various methods for estimating VO_2_ during physical activity, the use of the procedure validated in this study, which employs Equation (1) primarily based on netHR, is not only innovative but also offers several advantages. It is a straightforward procedure accessible to all practitioners, utilizing HR values that are currently easily obtainable through affordable instruments. It does not require the knowledge or estimation of the subject’s HR_max_, and it incorporates HR_rest_, a characteristic unique to each individual and highly responsive to training. The presented formula (Equation (1)), while suitable for estimating netVO_2_ (above the resting VO_2_), can also serve as a basis for estimating VO_2_ during any moderate-to-long-duration activity of submaximal intensity. To do so, the resting VO_2_ value must be added, which is commonly assumed to be 3.5 mL/(kg·min) [7].

From a practical standpoint, the knowledge of the VO_2_ associated with a specific training session can be of great value for estimating EE in that activity. That is, understanding EE, for example, in kcal, is crucial if an individual aims to control caloric intake for weight gain or loss. On the other hand, evaluating VO_2_ and, by extension, evaluating EE during exercise can play a significant role in improving and understanding people’s health. It is possible to evaluate how the cardiovascular capacity has changed over time by knowing the VO_2_ linked to a specific workout. The effectiveness of the respiratory and circulatory systems in getting oxygen into the muscles is also reflected in VO_2_. A lower risk of cardiovascular and respiratory disorders as well as an enhanced quality of life are linked to a good aerobic capacity [33,34]. In addition, the estimation of EE during physical activity yields data regarding the approximate caloric expenditure during physical activity. Maintaining a balance between energy intake and expenditure is critical for maintaining health and body weight because it prevents unintended weight gain and lowers the risk of obesity, type 2 diabetes, and other related illnesses [35,36]. From the standpoint of exercise prescription, a precise understanding of intensity levels enables trainers and medical practitioners to tailor exercise regimens to the specific needs and objectives of every person. In this context, the use of heart rate in estimating VO_2_ and EE must consider other variables to increase the accuracy of the estimation. In our study, we found that netHR supplemented by “sex” and “BMI” allows for a valid estimation.

Using VO_2_ values, EE can be easily estimated, making it relatively accessible to all practitioners and trainers. Based on older knowledge that led to the development of the “R table”, it is known that for every liter of consumed oxygen, our body can produce energy ranging from 4686 kcal to 5047 kcal [1]. This variation depends on the mix of nutrients metabolized during physical activity, which determines the respiratory quotient (R), which can range between 0.707 and 1 [12]. A lower value results indicates almost exclusive fat degradation, while a higher value indicates exclusive carbohydrate degradation. 

Since the degradation of fatty acids in the citric acid cycle continues only if sufficient oxaloacetate and other intermediates from carbohydrate breakdown processes combine with the acetyl-CoA formed during β-oxidation [1], lower R values are rarely encountered. In this context, we assume that R (respiratory quotient) reflects the value of the respiratory exchange ratio measured using pulmonary ventilation. This similarity makes sense if the assessment is made during stabilized intensity, sub-maximal, and continuous physical activities, as is the case here. A respiratory quotient equal to 1 is only possible in high-intensity activities. In moderate to vigorous activities, common in health-related physical activity, an R between 0.85 and 1 is typical. R values in this range imply that the “average” energy produced from one liter of consumed oxygen is approximately 5 kcal.

Considering the above, and with a valid estimate of VO_2_ from netHR (Equation (1)), estimating EE is relatively simple. It involves calculating the liters of oxygen expended in a particular activity and multiplying it by 5. For example, assume a male subject with a body weight of 70 kg, a BMI of 25, and a HR_rest_ of 65 bpm engages in a 1 h brisk walking session. The recorded average HR is 125 bpm, corresponding to a netHR of 60 bpm (netHR = 125 − 65). Substituting these values into the equation,
Average netVO_2_ (mL/(kg·min)) during the training session = 16 + 3.619 + (0.269 × 60) − 0.569 × 25 = 21.53 mL/(kg·min)
Total VO_2_ during the session = 21.53 mL/(kg·min) + Resting VO_2_ (3.5 mL/(kg·min)) = 25.03 mL/(kg·min)
Total liters of VO_2_ in the session = 25.03 (mL/(kg·min)) × 60 (min) × 70 (kg)/1000 = 105.13 L
Estimated EE = 105.13 × 5 = 525.7 kcal

Another practical application of estimating VO_2_ during a training session is determining the level of effort intensity in MET (metabolic equivalent of task). A MET value of 1 represents the energy expenditure at rest, which serves as a basis for determining the exercise intensity by comparing the energy expenditure during the activity with the resting value. Although MET values are often expressed in EE units (joules or kcal), it is widely accepted that 1 MET is equivalent to 3.5 mL/(kg·min) of oxygen consumption. Therefore, for every 3.5 mL/(kg·min) of oxygen consumption, there is a 1 MET increase in effort intensity. This allows for the assessment of effort intensity in any training session or the prescription of intensity levels in MET.

Referring to the previous example, knowing that the average total VO_2_ during a one-hour training session is 25.03 mL/(kg·min), dividing this value by 3.5 reveals the session’s intensity in MET.

In this case, it corresponds to 7 MET. A MET intensity value of 7 falls within the “vigorous” physical activity range, according to the ACSM intensity classification [7]. Therefore, in the same individual, netHR can be used both to determine the intensity of the physical activity and to prescribe levels of physical activity intensity based on “MET” values. In our view, the validated procedure in this study, which estimates netVO_2_ mainly from netHR, can be highly useful in various exercise assessment and prescription contexts. It can also be used to estimate EE universally in a simple and cost-effective manner. Currently, various affordable portable electronic devices are available on the market that allow for easy and accurate HR measurements.

Some limitations should be considered. This procedure estimates the VO_2_ and does not directly evaluate respiratory exchanges. In addition, the VO_2_ estimation can only be used in predominantly aerobic activities of moderate to long duration, such as those used in improving cardiovascular capacity. The results were obtained for continuous tasks, making this procedure unsuitable for intermittent tasks. The validation in our study was achieved using specific tests and types of exercise; it was carried out on a sample with the characteristics mentioned above. It is not intended to validate the procedure for all types of exercise or for all population groups.

## 5. Conclusions

The netHR, in combination with sex and BMI, can be used to estimate NetVO_2_ in some types of exercises that are used to improve cardiovascular capacity in active adult subjects. From here, it is possible to estimate VO_2_ in any activity by simply adding the resting VO_2_ value, typically assumed to be 3.5 mL/(kg·min) (1 MET). The validation process revealed non-significant differences with a high level of agreement between the measured and predicted values. When applied to the same individual, this netVO_2_ estimation procedure relies almost exclusively on netHR. Additionally, it was found that it is possible to calculate the EE resulting from a specific training session, and to determine or prescribe the exercise intensity in MET. From a practical standpoint, in cardiovascular capacity training sessions, the estimation of VO_2_ and EE can be performed in a simple manner, which is accessible to any healthy individual.

## 6. Practical Applications

The procedure for estimating VO_2_ and EE in activities such as walking/running, cycling, or climbing up/down steps based on netHR is one of the most accessible and easy to use procedures. Indirectly, it makes it possible to assess or prescribe an exercise intensity in MET values. Using this procedure does not require knowledge of maximum values such as HR_max_ or VO_2__max_. It is a process that considers specific individual variables, such as HR_rest_.

The simplified equation allows the oxygen consumption above the resting value to be estimated as follows: ${netVO}_{2}\;(mL/(kg\cdot min)=16+3.6\cdot \left(Sex\right)+0.3\cdot \left(netHR\right)-0.6\cdot \left(BMI\right).\;$If we consider VO_2rest_ (1 MET) to be equal to 3.5 mL/(kg·min), the average total oxygen consumption in a given activity can be obtained by adding this value.

## Figures and Tables

**Figure 1 jfmk-09-00066-f001:**
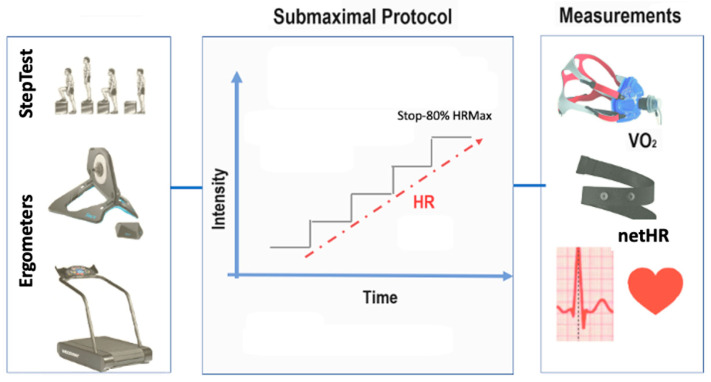
Schematic for data collection procedures.

**Figure 2 jfmk-09-00066-f002:**
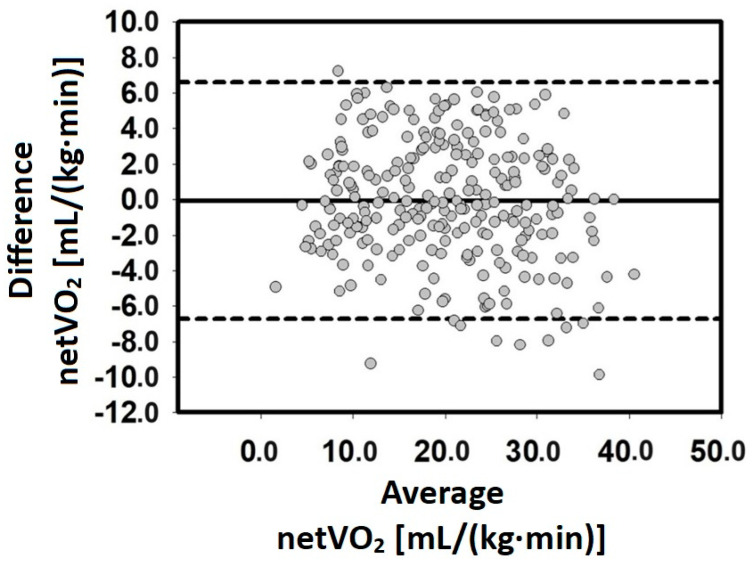
Bland–Altman plots of the measured and estimated netVO_2_. The y-axis represents the difference between the measured and estimated netVO_2_ [mL/(kg·min)]. The x-axis represents the average between the measured and estimated netVO_2_ [mL/(kg·min)]. These data are from the validation group.

**Table 1 jfmk-09-00066-t001:** General demographic characteristics of the entire sample (SD—standard deviation).

	N	Minimum	Maximum	Mean	SD
Age [years]	156	18	81.0	42.0	20.7
Body mass [kg]	156	47.8	116.0	73.1	11.7
Height [cm]	156	138	190	169	11
BMI [kg/m^2^]	156	17.7	44.3	25.9	4.3
HR_rest_ [bpm]	156	49	110	68	11
VO_2rest_ [mL/(kg·min)]	156	1.2	6.4	3.2	0.9

BMI—body mass index; HR_rest_—resting heart rate; VO_2rest_—resting oxygen consumption.

**Table 2 jfmk-09-00066-t002:** Inferential statistics that show the absence of statistically significant differences between the two groups (group 1: prediction; group 2: validation) in the different variables evaluated.

	Group	Mean	SD	*t*-Test (*p*-Value)
Age [years]	1 (n = 79) 2 (n = 77)	41.6 42.3	20.4 21.1	−0.20 (0.841)
Body mass [kg]	1 (n = 79) 2 (n = 77)	73.5 72.8	11.6 11.8	0.360 (0.719)
Height [cm]	1 (n = 79) 2 (n = 77)	169 167	12 11	0.535 (0.593)
BMI [kg/m^2^]	1 (n = 79) 2 (n = 77)	25.8 26.0	4.0 4.7	−0.186 (0.853)
HR_rest_ [bpm]	1 (n = 79) 2 (n = 77)	69 67	11 11	0.464 (0.644)
VO_2rest_ [mL/(kg·min)]	1 (n = 79) 2 (n = 77)	3.3 3.2	1.0 0.8	0.227 (0.821)

BMI—body mass index; HR_rest_—resting heart rate; VO_2rest_—resting oxygen consumption.

**Table 3 jfmk-09-00066-t003:** *t*-test paired samples comparison between the measured and estimated VO_2_ in the validation group. Effect size index (Cohen’s d) is also presented.

Measured netVO_2_ [mL/(kg·min)]	Estimated netVO_2_ [mL/(kg·min)]			
Mean ± 1SD	Mean ± 1SD	Mean Difference (95% CI)	*t*-Test (*p*-Value)	d [Descriptor]
20.09 ± 8.91	20.29 ± 8.41	0.0568 (−0.36 to 0.47)	0.27 (0.788)	0.01 [trivial]

netVO_2_—net oxygen uptake.

## Data Availability

No new data were created or analyzed in this study. Data sharing is not applicable to this article.

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
