# Peer review of "Net Heart Rate for Estimating Oxygen Consumption in Active Adults"

_jfmk, 2024, doi:10.3390/jfmk9020066_

Round 1

Reviewer 1 Report (Previous Reviewer 1)

Comments and Suggestions for Authors

Thanks for addressing my comments and suggestions. I still have comments.

You have two different descriptions of net heart rate. Your description for net heart rate (L13) is “the magnitude of the variation in heart rate”. That is incorrect. If your heart rate e.g. is 160, 164 and 162, then it is the variation in those numbers. Is net heart not just an absolute value so for if exercise is 160 and rest is 60, your net heart rate is 100. Please clarify your description if needed. This is in contrast with L23 “netHR is the amount of heart rate (beats per minute) above rest”. Please clarify. See also L48.

L52. “netHR fluctuation to VO2 variation”. Do you mean “netHR to VO2” or is the variation of both variables. The reviewer finds your writing imprecise and inconsistent. Please revise.

Table 2. Change “26,0” to “26.0”.

L91. “the total number of plots”. Do you mean “the total number of measurements” or the “the total number of datapoints”. Please reconsider whether plot is the right time here and throughout the manuscript. For example L174 “80% of the plots”. Use of plot in Bland-Altman plot is fine.

L113. I suggest to delete “For some reason, it might be necessary to stop the test due to an unpredictable situation.”

L129. “pleasant environment”. I suggest to delete or clarify as it is vague. For example, were the lights dimmed, no use of electronic devices etc.

L175. “(95CI%)” can be deleted. It seems you state 80% within a 95% confidence interval. I am not sure that is right.

Ls 179-180. Please provide the BMI data that show the change of BMI for the individuals in the study as you mention this in the results section. Please update your methods that BMI was measured over a period of time in the study.

Table 3. “20.086 ± 8.91 20.294 ± 8.41”. Is this precision with 3 decimal places allowed?

L212. Is the Ref 24 in Spanish needed. In addition, the respiratory quotient is reserved for the measurement at cell level. Maybe better to use respiratory exchange ratio.

L220. Ref 25 is in Spanish. It should be possible to replace with a references in English. Use only Spanish references when there is no English one available. You seem to refer here to general facts.

L222. Ref 32 is review article. Please make that clear when a review article is used, e.g. “for a review see [32].

L222. Ref 33 is on heart rate variability. Please clarify.

L223. Ref 34 is not on exercise but chronic stress in office workers. Please replace.

Ls 261-262. Is fat degradation the same as fat breakdown? Are your talking about “fat oxidation”. Please be consistent in terminology.

Ls 290-291. “it is widely accepted that 1 MET is equivalent to 3.5 mL/(kg·min) of oxygen consumption”. Not sure whether that is the case, see DOI: 10.1152/japplphysiol.00023.2004 and DOI: 10.1097/01.HCR.0000270693.16882.d9

L321. “accessible to any individual.” Are you sure? Is it for people with COPD? Please be precise.

Comments on the Quality of English Language

Manuscript has improved but has still issues. 

Author Response

REVIEWER #1

Comments and Suggestions for Authors

Thanks for addressing my comments and suggestions. I still have comments.

Authors: Thank you very much for the time you spent and your constructive feedback on this manuscript. We have made every effort to take on board your recommendations and comments. We hope this revised version and the responses to the comments (kindly refer to our replies below) will meet your requirements. Please note that all new changes in the revised manuscript are highlighted in yellow.

You have two different descriptions of net heart rate. Your description for net heart rate (L13) is “the magnitude of the variation in heart rate”. That is incorrect. If your heart rate e.g. is 160, 164 and 162, then it is the variation in those numbers. Is net heart not just an absolute value so for if exercise is 160 and rest is 60, your net heart rate is 100. Please clarify your description if needed. This is in contrast with L23 “netHR is the amount of heart rate (beats per minute) above rest”. Please clarify. See also L48.

Authors: We appreciate the reviewer’s advice. We have altered the manuscript (shaded in yellow) to clarify the concept of netHR.

L52. “netHR fluctuation to VO2 variation”. Do you mean “netHR to VO2” or is the variation of both variables. The reviewer finds your writing imprecise and inconsistent. Please revise.

Authors: We changed the manuscript to be more precise about what we wanted to say. In this case, we were referring to the variation of both variables.

Table 2. Change “26,0” to “26.0”.

Authors: We appreciate the reviewer’s advice. This was edited accordingly.

L91. “the total number of plots”. Do you mean “the total number of measurements” or the “the total number of datapoints”. Please reconsider whether plot is the right time here and throughout the manuscript. For example L174 “80% of the plots”. Use of plot in Bland-Altman plot is fine.

Authors: What we meant to say was "data points". We have changed the text accordingly.

L113. I suggest to delete “For some reason, it might be necessary to stop the test due to an unpredictable situation.”

Authors: We have deleted that sentence as advised.

L129. “pleasant environment”. I suggest to delete or clarify as it is vague. For example, were the lights dimmed, no use of electronic devices etc.

Authors: We have changed the text accordingly as advised for clarity’s sake.

L175. “(95CI%)” can be deleted. It seems you state 80% within a 95% confidence interval. I am not sure that is right.

Authors: We understand the reviewer’s comment. This was deleted as advised.

Ls 179-180. Please provide the BMI data that show the change of BMI for the individuals in the study as you mention this in the results section. Please update your methods that BMI was measured over a period of time in the study.

Authors: This is a cross-sectional study. Participants were only assessed once. What we meant to say (when referring: "As for the same individual, gender remains constant and BMI changes slowly over de course of time (weeks or months), we observed that the estimation of netVO2 is almost exclusively linked to netHR.") is that if netVO2 is estimated in the same individual periodically over time, the variable that most determines the result is netHR. Because in the same individual, gender doesn't change, and BMI doesn't usually change markedly over a few weeks or even months.

However, we agree that this phrase should not be placed in the results, but in the discussion.

We have removed this sentence from the results. Its content was already mentioned in the Discussion.

Table 3. “20.086 ± 8.91 20.294 ± 8.41”. Is this precision with 3 decimal places allowed?

Authors: We understand the reviewer’s comment. Two decimals were used now.

L212. Is the Ref 24 in Spanish needed. In addition, the respiratory quotient is reserved for the measurement at cell level. Maybe better to use respiratory exchange ratio.

Authors:

We have clarified our intention in the manuscript (lines 282-287)

L220. Ref 25 is in Spanish. It should be possible to replace with a reference in English. Use only Spanish references when there is no English one available. You seem to refer here to general facts.

Authors: We have changed the reference of the book in question to the English version.

L222. Ref 32 is review article. Please make that clear when a review article is used, e.g. “for a review see [32].

Authors: We have altered de text accordingly.

L222. Ref 33 is on heart rate variability. Please clarify.

Authors: Thank you very much for your thoughtful comment. This reference was not the correct one. We have changed it to the correct reference.

L223. Ref 34 is not on exercise but chronic stress in office workers. Please replace.

Authors: We have removed this reference as advised.

Ls 261-262. Is fat degradation the same as fat breakdown? Are your talking about “fat oxidation”. Please be consistent in terminology.

Authors: We want to say: “degradation of fatty acids in the citric acid cycle continues only if sufficient oxaloacetate and other intermediates from carbohydrate breakdown process combine with the acetyl-CoA formed during β-oxidation”.

We changed de text to clarify.

Ls 290-291. “it is widely accepted that 1 MET is equivalent to 3.5 mL/(kg·min) of oxygen consumption”. Not sure whether that is the case, see DOI: 10.1152/japplphysiol.00023.2004 and DOI: 10.1097/01.HCR.0000270693.16882.d9

Authors: The question is very pertinent and has been the subject of our reflection. There are indeed several studies that did not find 1MET values very close to 3.5 ml/kg/min in their respective samples.

In previous studies, we ourselves have measured different "resting" VO2 values; in our case, in elderly people, we found resting VO2 values close to “3”.

In this study, as we can see in table 2, VO2rest (1MET) is 3.3 in group 1; and it is 3.2 in G2.

Since the values in different studies are not precisely "3.5", and the "n" of the samples is not very high, it seems appropriate to use the value 3.5 as indicated by the ACSM.

However, we have to bear in mind that to really know the value of 1MET we would have to measure VO2 individually in every person, which is impossible. Therefore, the value of 3.5 ml/kg/min as the equivalent of oxygen consumption associated with 1MET is the most commonly used. We must also bear in mind that this value is only used to estimate energy expenditure, with advantages and disadvantages. One disadvantage is that it's not very accurate; the big advantage is that it can be used on a large scale.

L321. “accessible to any individual.” Are you sure? Is it for people with COPD? Please be precise.

Authors: When we say that the test is accessible to everyone, we mean everyone who has the minimum conditions to take it and who doesn't have any health problems or other conditions that would prevent them from taking it. But we agree with the suggestion.

We have changed the sentence in the text to : "… o any healthy individual.

Comments on the Quality of English Language

Manuscript has improved but has still issues.

Authors: We made our best to improve the manuscript. Nonetheless, please note that if there are proofread issues, these will be correct during the proofs.

Reviewer 2 Report (New Reviewer)

Comments and Suggestions for Authors

NET HEART RATE FOR ESTIMATING OXYGEN CONSUMPTION IN ACTIVE ADULTS

General Commentary

This article presents a very interesting and pertinent question of research on verify the accuracy of net heart rate (netHR) as a method for calculating oxygen consumption (VO2) in active adults.

However, some questions need to be clarified in order to better understand and apply the results found.

MODERATE CONSIDERATION

FULL TEXT

What is the justification for the text being marked in yellow? Is this related to the response letter? Where is the response letter if applicable?

INTRODUTION

Objective

I believe that the objective failed to highlight the tests that were used for such estimates, such as maximum tests on cycle ergometer, treadmill, among others.

METHODS

Study Design

Please initially add a subchapter on the study design to the methods, with an illustrative time-line figure.

Data Analysis

In relation to data analysis of power values, speed, heart rate, oxygen consumption. At no point in the methods was it reported how the data was analyzed, what type of filters were used? How, for example, was VO2max defined? among other values calculated by the authors for subsequent statistical analysis.

Therefore, I suggest adding an entire subchapter in the methods titled Data Analysis.

Statistical

In relation to the regression model adopted, why was this model excluded instead of, for example, stepwise? Furthermore, the authors tested cubic, quadratic or polynomial regression models to actually verify the certainty of the model used. Since they are proposing a validation process?

In relation to the term validation, I think it is inappropriate and I suggest that prediction or determination be used. The process of validating something requires many more evaluations, with different populations among other factors not investigated in this study.

RESULTS AND DISCUSSION

Adjust results and discussion based on the comments mentioned above in the statistics if necessary

PRACTICAL APPLICATION

Please insert final chapter (Practical Application).

Author Response

REVIEWER #2

Comments and Suggestions for Authors

NET HEART RATE FOR ESTIMATING OXYGEN CONSUMPTION IN ACTIVE ADULTS

General Commentary

This article presents a very interesting and pertinent question of research on verify the accuracy of net heart rate (netHR) as a method for calculating oxygen consumption (VO2) in active adults. However, some questions need to be clarified in order to better understand and apply the results found.

Authors: Thank you very much for the time you spent and your constructive feedback on this manuscript. We have made every effort to take on board your recommendations and comments. We hope this revised version and the responses to the comments (kindly refer to our replies below) will meet your requirements. Please note that all new changes in the revised manuscript are highlighted in yellow.

MODERATE CONSIDERATION

FULL TEXT

What is the justification for the text being marked in yellow? Is this related to the response letter? Where is the response letter if applicable?

Authors: We think it was an oversight. The text should have been sent to you unmarked. As aforementioned, only new changes are highlighted in yellow.

INTRODUTION

Objective

I believe that the objective failed to highlight the tests that were used for such estimates, such as maximum tests on cycle ergometer, treadmill, among others.

Authors: We agree that the types of exercise used should be mentioned when defining the objectives. We have amended accordingly.

METHODS

Study Design

Please initially add a subchapter on the study design to the methods, with an illustrative time-line figure.

Authors: We have introduced a short sub-chapter as suggested.

Data Analysis

In relation to data analysis of power values, speed, heart rate, oxygen consumption. At no point in the methods was it reported how the data was analyzed, what type of filters were used? How, for example, was VO2max defined? among other values calculated by the authors for subsequent statistical analysis. Therefore, I suggest adding an entire subchapter in the methods titled Data Analysis.

Authors:  As mentioned in the text, all the protocols are submaximal. VO2 max is therefore not measured. HRmax is not measured either. One of the advantages of using netHR is precisely that it is not necessary to know either VO2max or HRmax.

Each protocol is described individually.

However, we feel that we can clarify the methodology used more clearly. More specifically, we can say that the data collected (VO2 and HR) was treated individually, by intensity level. Each participant only performed one protocol. Therefore, for each subject, several data points were taken into account; as many as the levels performed.

We have added some details to the text in order to clarify the methodology used.

Statistical

In relation to the regression model adopted, why was this model excluded instead of, for example, stepwise? Furthermore, the authors tested cubic, quadratic or polynomial regression models to actually verify the certainty of the model used. Since they are proposing a validation process?

Authors: We understand the reviewer’s comment. However, researchers and statisticians have identified numerous statistical problems with the “stepwise” method which includes overfitting the data, biased estimates, and inflated Type I error (Harrell, Jr, F. E., & Harrell, F. E. (2015). General aspects of fitting regression models. Regression modeling strategies: with applications to linear models, logistic and ordinal regression, and survival analysis, 13-44).

Conversely, backward elimination is a technique that starts with a full set of features and iteratively removes one feature at a time based on a predefined criterion. The aim is to eliminate the least informative feature(s) at each step, gradually refining the feature set. This technique more suitable because it can help to reduce the chances of overfitting the data and make the linear regression model more interpretable. This method has been widely used, inclusively by one lead researcher in statistics that for predictive models uses linear rather than polynomial regressions (Alan M Nevill, e.g., Nevill, A. M., Negra, Y., Myers, T. D., Sammoud, S., & Chaabene, H. (2020). Key somatic variables associated with, and differences between the 4 swimming strokes. Journal of Sports Sciences, 38(7), 787-794; Nevill, A. M., Bate, S., & Holder, R. L. (2005). Modeling physiological and anthropometric variables known to vary with body size and other confounding variables. American Journal of Physical Anthropology: The Official Publication of the American Association of Physical Anthropologists, 128(S41), 141-153; Burrows, M., Nevill, A. M., Bird, S., & Simpson, D. (2003). Physiological factors associated with low bone mineral density in female endurance runners. British journal of sports medicine, 37(1), 67-71).

In relation to the term validation, I think it is inappropriate and I suggest that prediction or determination be used. The process of validating something requires many more evaluations, with different populations among other factors not investigated in this study.

Authors: We understand and agree with the reviewer. Minor editing was done for clarity’s sake.

RESULTS AND DISCUSSION

Adjust results and discussion based on the comments mentioned above in the statistics if necessary

Authors: From the discussion, aspects were added that were in line with what had been suggested. Limitations of the study related to the reviewer's suggestions were also added.

PRACTICAL APPLICATION

Please insert final chapter (Practical Application).

Authors: We added the chapter “Practical applications” as suggested.

Round 2

Reviewer 1 Report (Previous Reviewer 1)

Comments and Suggestions for Authors

Thanks for addressing all my comments. There is still some misunderstanding.

L50. You state “the use of net heart rate (netHR), or the magnitude of HR variation above resting (HRrest) measured during physical activity or exercise,…”

There is still inconsistency with what netHR is as the description of netHR in the abstract is different.

Please revise and ensure consistency of the description of netHR.

The “magnitude of HR variation” is not describing the heart rate value above rest.

Your netHR is the magnitude of HR above the resting value.

Table 3. When you have two decimal places for the absolute values, you should also express the mean difference and 95%CI with two decimal places.

Figure 2. Change “axes” to “axis”.

Author Response

REVIEWER #1

Thanks for addressing all my comments. There is still some misunderstanding.

Authors: Thank you very much for the time you spent and your constructive feedback on this manuscript. We have made every effort to take on board your recommendations and comments. We hope this revised version and the responses to the comments (kindly refer to our replies below) will meet your requirements. Please note that all new changes in the revised manuscript are highlighted in yellow.

L50. You state “the use of net heart rate (netHR), or the magnitude of HR variation above resting (HRrest) measured during physical activity or exercise,…”

Authors:

We have amended the text to make our intention clearer and to define the concept more objectively.

There is still inconsistency with what netHR is as the description of netHR in the abstract is different.

Authors:

We have amended the text to be consistent with what is stated in the abstract.

Please revise and ensure consistency of the description of netHR.

Authors:

We have amended the text to clarify.

The “magnitude of HR variation” is not describing the heart rate value above rest.

Authors:

We have amended the text to clarify.

Your netHR is the magnitude of HR above the resting value.

Authors:

Yes. We clarified it in the text

Table 3. When you have two decimal places for the absolute values, you should also express the mean difference and 95%CI with two decimal places.

Authors:

We have changed accordingly

Figure 2. Change “axes” to “axis”.

Authors:

We have changed accordingly

This manuscript is a resubmission of an earlier submission. The following is a list of the peer review reports and author responses from that submission.

Round 1

Reviewer 1 Report

Comments and Suggestions for Authors

Title indicates “active adult” and abstract “physically active people”. Please be consistent considering the cohort of the study.

The title indicates just focus on net heart rate but sex and BMI are part of the equation as well. Please revise the title.

Throughout the manuscript, please change “mL/kg/min” to “mL·kg-1·min-1” or “mL/(kg·min)” and ensure you use middle dots for multiplication.

The description the treadmill and cycling protocol seem to have stage duration in which steady-state could have been obtained but the step test protocol has changes every minute. Why was the stage duration different between the protocols.

Ls 22-23. I suggest to start the abstract with description of net heart rate and net oxygen consumption.

L24. Please revise “vo2”, seems to be a font issue.

L24. I suggest to consider https://journals.physiology.org/doi/epdf/10.1152/japplphysiol.00353.2022 and use this reference in the discussion. See also L53.

L34. I suggest to reconsider whether it is better to use “steady-state”. Stabilized is not common. In addition, replace reference 1 with an English language source.

Ls 37-38. Replace “used. [2–6].” with “used [2–6].”

L40. Is the reserve heart rate the same as the net heart rate. Please clarify.

L46. The description of net heart rate seems to be imprecise as it mentions the fluctuation indicates a dynamic situation where is a change of values. Please reconsider the description.

L57. I suggest to delete “The HR does not reach the value of zero when one is at rest”. I am sure your audience will not surprised.

Ls 62-63. Please provide a reference to support this statement.

L68. Please rephrase “in the utter inactivity”.

L73. “and other variables”.  Please be precise. In addition, the introduction needs to provide justification why these other variables were considered. For example, BMI is mentioned but why BMI and lean body mass or body fat%.

Table 1. Height should be cm. Please present height and heart rate values without decimal places. For other parameters, be consistent with the number of decimal places for each parameter. Same comment for Table 2 parameters and values.

L90. It looks that allocation and not randomization took place.

L110. “until the subject requested to stop the test” How often did that happen?

Figure 1. A step test picture is not a picture of an ergometer. Please change.

Figure 1. What is “HRRecovery”?

L125. Please clarify what was done to have the environment experienced as pleasant.

L135. “or if the subject asked for it to end”. How often did this happen? See L141 as well.

L151. Why was the heart rate measured during recovery. That data is not present in the paper.

L172. Please clarify “meaningfully”. See also L197.

L175. “and BMI changes slowly over time”. How it that possible in a session? Please provide that data to support this statement.

Table 3. Is this accuracy of 3 decimal places allowed? Please justify.

Throughout the manuscript, please change “95CI” to “95CI%”

L190. “. Specifically, nearly all plots were within the 95CI”. Do you mean datapoints?

Figure 2. The labels for the Bland-Altman plot are not common. Please be specific.

L206. Ref 27 is using the %HRR. Please clarify?

L126. Equation 1 has not only netHR. Please revise.

L252 “Given that fat degradation is part of the glycolysis metabolism pathway”. Please revise. The reviewer is actually quite shocked that such a statement violating knowledge on energy metabolism is present in the manuscript.

L282. “A value of 7 MET intensity falls within the "moderate" physical activity range, according to the ACSM intensity classification [7].” That is not correct, see e.g.  DOI: 10.1093/eurpub/ckac078 and https://doi.org/10.1136/bjsports-2020-102955 and DOI: 10.1249/MSS.0b013e318213fefb

References. No PMID info is required. See ref 2 e.g.. Needs revision.

Reviewer 2 Report

Comments and Suggestions for Authors

In the introduction, authors must cite and reference works that already exist in the literature that estimate VO2max from HR and thus find a gap that justifies study.

The authors do not explain in the methodology how they calculated Net Heart rate

The authors should clarify why the progressive test at 80%Hr should be interrupted when they could obtain Hrmax and VO2max data

There is a major limitation in the use and obtaining of data, and little transparency regarding data acquisition, which limits its interpretation.
